# Impact of targeted diabetic retinopathy training for graders in Vietnam and the implications for future diabetic retinopathy screening programmes: a diagnostic test accuracy study

Katie Curran [1], Nathan Congdon,[1,2,3] Tung Thanh Hoang,[4,5] Lynne Lohfeld,[1,6] Van Thu Nguyen,[7] Hue Thi Nguyen,[7] Quan Nhu Nguyen,[8] Catherine Dardis,[9] Gianni Virgili [1,9] Prabhath Piyasena [1] Huong Tran,[7] Recivall Pascual Salongcay,[1] Mai Quoc Tung,[8] Tunde Peto [1,9]

For numbered affiliations see end of article.

**Correspondence to**
Dr Katie Curran;
K.Curran@qub.ac.uk

## ABSTRACT

**Objectives** To compare the accuracy of trained level 1 diabetic retinopathy (DR) graders (nurses, endocrinologists and one general practitioner), level 2 graders (midlevel ophthalmologists) and level 3 graders (senior ophthalmologists) in Vietnam against a reference standard from the UK and assess the impact of supplementary targeted grader training.

**Design** Diagnostic test accuracy study.

**Setting** Secondary care hospitals in Southern Vietnam.

**Participants** DR training was delivered to Vietnamese graders in February 2018 by National Health Service (NHS) UK graders. Two-field retinal images (412 patient images) were graded by 14 trained graders in Vietnam between August and October 2018 and then regraded retrospectively by an NHS-certified reference standard UK optometrist (phase I). Further DR training based on phase I results was delivered to graders in November 2019. After training, a randomised subset of images from January to October 2020 (115 patient images) was graded by six of the original cohort (phase II). The reference grader regraded all images from phase I and II retrospectively in masked fashion.

**Primary and secondary outcome measures** Sensitivity was calculated at the two different time points, and $\chi^2$ was used to test significance.

**Results** In phase I, the sensitivity for detecting any DR for all grader groups in Vietnam was low (41.8–42.2%) and improved in phase II after additional training was delivered (51.3–87.2%). The greatest improvement was seen among level 1 graders (p<0.001), and the lowest improvement was observed among level 3 graders (p=0.326). There was a statistically significant improvement in sensitivity for detecting referable DR and referable diabetic macular oedema between all grader levels. The post-training values ranged from 40.0 to 61.5% (including ungradable images) and 55.6%–90.0% (excluding ungradable images).

**Conclusions** This study demonstrates that targeted training interventions can improve accuracy of DR grading. These findings have important implications for improving

## STRENGTHS AND LIMITATIONS OF THIS STUDY

⇒ Graders in Vietnam were trained to detect diabetic retinopathy (DR) based on the UK's DR screening model.
⇒ This study describes the impact of a training intervention to improve DR grading in Vietnam.
⇒ Gradable and ungradable fundus image grading were included in the analysis.
⇒ The sample size was smaller in phase II compared with phase I.

service delivery in DR screening programmes in low-resource settings.

## INTRODUCTION

The prevalence of diabetes among adults in Vietnam is approximately 6% and has almost doubled over the past decade.[1] Early detection through diabetic eye screening programmes (DESPs) is important to reduce the risk of avoidable blindness due to diabetic retinopathy (DR). Since the introduction of systematic DESPs in the UK, a high-income country, DR is no longer the leading cause of blindness among working age adults.[2] The key to such successful DESPs is implementing accurate, innovative and cost-effective models tailored to fit healthcare systems and contexts.

Investing in training personnel to increase human resources and procuring appropriate diagnostic and treatment equipment are essential to ensure that service providers can deliver optimum care for people with DR. In low-middle income countries (LMICs), there is often insufficient capacity to implement robust DESPs due to the lack of skilled human resources and infrastructure.[3 4] In

**BMJ**

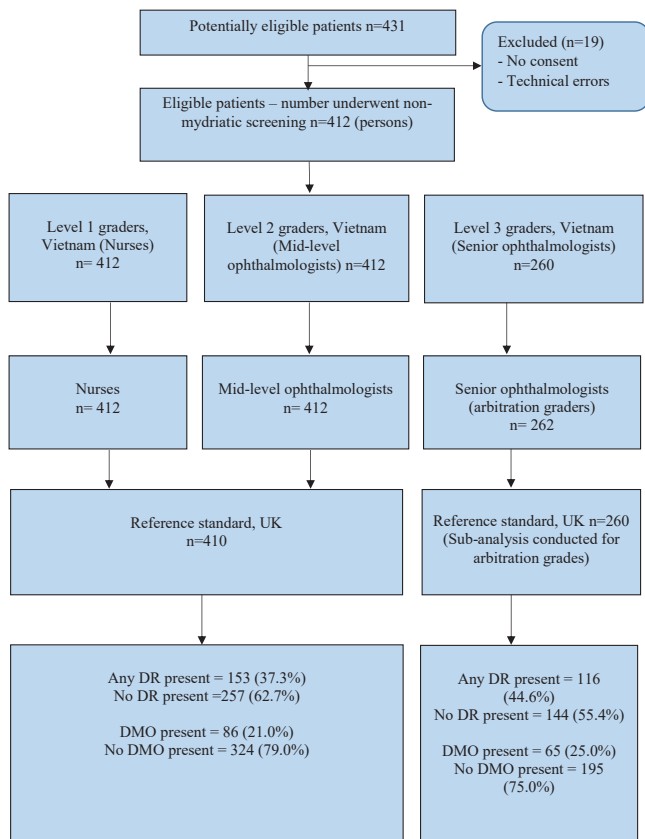

**Figure 1** Flow diagram to illustrate enrolment of patients and management of images in phase I from August to October 2018 (initial grading performance analysis). Level 1 and level 2 graders graded the same set of photographs and level 3 graders graded a subset of these photographs: all disagreements between level 1 and 2 graders and a 40% random sample of all images. DMO, diabetic macular oedema; DR, diabetic retinopathy.

Vietnam, there are only 14 ophthalmologists per million population compared with 49 per million in the UK.[5]

All screening programmes must provide evidence of their ability to detect the targeted condition and ensure that the service performs efficiently to improve screening accuracy when it falls short. To date, there is insufficient evidence on DR grading accuracy using non-mydriatic digital imaging by trained graders in LMICs and even less about the capacity of DESPs in LMICs to improve where poor accuracy is detected. The current retrospective study is designed to assess accuracy of a range of graders in a non-governmental organisation (NGO) supported DESP in Vietnam and to study the efficacy of a quality improvement intervention.

## METHODS
### Study participants
The 14 participants from Vietnam in phase I included: level 1 DR graders (six nurses, one general practitioner and two endocrinologists, all with <1 year grading experience, 55.6% female), level 2 DR graders (three newly qualified

ophthalmologists with <1 year formal DR grading experience, 100% female) and level three DR graders (two senior ophthalmologists with >5 years' experience providing treatment for sight threatening DR, but with <1 year formal DR grading experience, 100% male). In phase II, 6/14 graders (three level 1, two level 2, one level 3) from phase I were included. The reference standard from the UK (KC) was a fully qualified optometrist trained in DR grading and certified by the UK NHS DESP.[6] Vietnamese level 1, 2 and 3 graders are equivalent to primary, secondary and arbitration graders, respectively, in UK DESPs.[7] In the current study, Vietnamese level 1 and level 2 graders graded all fundus images for DR. All images having disagreement between graders, and an additional randomly selected 40% of all images, were sent for arbitration grading by level 3 graders in Vietnam. All graders in Vietnam were masked to any prior diagnoses or grades of the reference standard, while the reference standard was also masked to results of grading in Vietnam. Fundus images were graded for 412 patients in phase I and 115 patients in phase II (figure 1 and figure 2).

### DR training for graders in Vietnam
As part of a DESP project supported by NGO Orbis International, a team of five Vietnamese doctors and medical administrators visited a Northern Ireland DESP in September 2017 to receive training on screening, programme administration and quality control methods. In February 2018, a senior UK NHS grader from the Belfast Trust (CD) and a fully qualified optometrist, trained in DR grading and certified by the NHS (KC), visited Vietnam to deliver DR training to graders involved in the DESPs (online supplemental figure S1 for training timeline). Training focused on ocular anatomy, retinal diseases, DR signs and grading (based on the UK National Screening Committee classification system) and appropriate referral pathways and management (online supplemental tables S1–S3).[8]

### Image acquisition and management
Images were captured by trained nurses and technicians in Vietnam. Two-field, 45° digital colour photographs (one disc centred and one macula centred) were taken using a tabletop non-mydriatic fundus camera (Canon CR2-AF, Canon Medical Systems Europe), in accordance with the UK's NHS DESP.[9] Nurses and technicians were trained to repeat inadequate images as a quality control measure and take anterior segment photographs where adequate fundus images were not possible. Images were anonymised and uploaded to a cloud-based software system (Spectra) for analysis by trained DR graders in Vietnam. The images were transferred to a Queen's University Belfast server for regrading by the reference standard.

### Assessment of gradability
Image quality was defined as 'adequate' or 'inadequate' in accordance with NHS DESP guidelines as outlined further:
▶ Adequate disc-centred image: complete optic disc >2 disc diameter (DD) from edge of image and fine vessels visible on surface of the disc.[9]

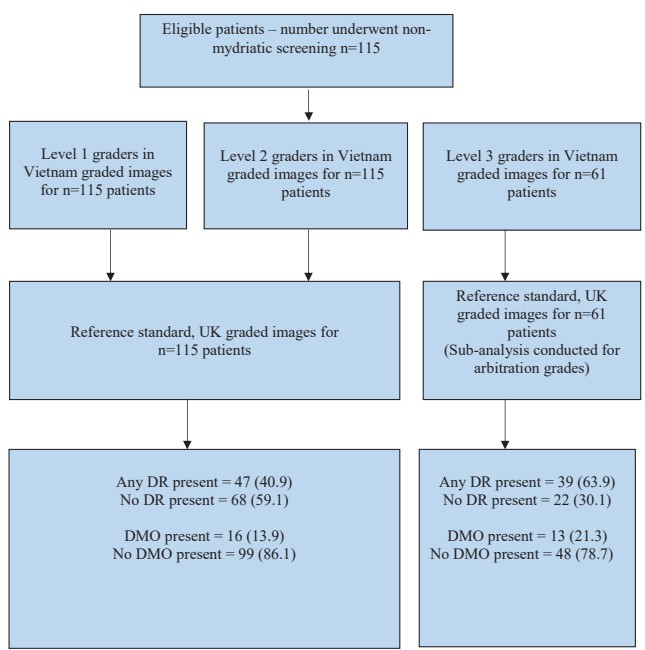

**Figure 2** Flow diagram illustrating the enrolment of patients and management of images included in phase II from January 2020 to October 2020 (follow-up grading performance analysis after retraining). Level 1 and level 2 graders graded the same set of photographs and level 3 graders graded a subset of these images. DMO, diabetic macular oedema; DR, diabetic retinopathy.

► Adequate macula-centred image: centre of fovea >2 DD from edge of image and vessels visible within 1DD of centre of fovea.[9]

The disc-centred and macula-centred images for each eye were viewed as a pair and graded at an individual eye level. The presence of DR and diabetic macular oedema (DMO) was also determined at a patient level and based on the worst affected eye. Participants with ungradable images were referred for further slit-lamp examination. Where images were considered inadequate but referable disease was detectable, the referable grade was recorded, and the patients were moved onto the appropriate referable grade pathway.[9]

Consecutive patients diagnosed with diabetes and undergoing evaluation for possible DR at Ho Chi Minh City General Hospital and Ho Chi Minh Eye Hospital (tertiary hospitals), Tien Giang General Hospital (provincial hospital) and Cai Ba General Hospital (district hospital) in Vietnam were recruited. Fundus images from August to October 2018 (phase I) were graded by 14 graders in Vietnam and then regraded retrospectively by a reference standard from the UK in phase I. Targeted remedial training, based on specific findings from the phase I analysis, was delivered in March 2019 and November 2019 by UK graders and Orbis (figure 1). Additionally, regular testing and training for quality assurance purposes was also introduced, similar to UK DESP

models. To evaluate the impact of this quality improvement intervention, a new subset of images was graded by six of the original cohort of graders between January and October 2020 (phase II) and regraded by the reference standard from the UK (KC) in September 2021.

### Statistical analysis

Data were entered into Microsoft Excel V.16.0 and then transferred to Stata V.17.0 (StataCorp LLC) for analysis. Intragrader and intergrader agreement was calculated using kappa, and a stratified random sampling technique was used to ensure a representative sample of images was regraded (online supplemental tables S4 and S5). Diagnostic test accuracy (DTA) comparing graders in Vietnam with the UK reference standard was assessed by calculating sensitivity, specificity, positive predicative values and negative predictive values. Sensitivity was calculated at the two different time points (phase I and phase II), and $\chi^2$ was used to test significance.

### Patient and public involvement

Patients or the public were not involved in the design, or conduct, or reporting, or dissemination plans of our research.

### RESULTS

### Patient characteristics

In phase I, 65.4% of patients were female with a mean age of 59.4 years. In phase II, 40.0% were female with a mean age of 59.8 years. Figures 1 and 2 describe enrolment of patients and capture and grading of images in phase I and II of the study, respectively.

### Initial grading performance analysis (phase I)

The sensitivity for detecting any DR was low against the reference standard in the UK for all grader groups in Vietnam. The sensitivity for detecting referable DR and referable DMO was even lower for all grader groups (table 1). The sensitivity increased when ungradable images were excluded from the analysis, though it still remained low (47.9–50.8% for any DR; 22.2%–38.1% for referable DR and 9.3–26.5% for referable DMO) (online supplemental table S6).

### Subsequent grading performance after retraining (phase II)

Subsequently, a further subset of images from 115 consecutive patients from January to October 2020 were graded by six of the original cohort of 14 Vietnamese graders and were regraded in the UK to evaluate graders' performance after targeted training was delivered, and quality control measures were instituted. The greatest improvement in sensitivity for detecting any DR was seen among level 1 graders (difference: +45.4%, 95% CI +33.1% to +57.8%; p<0.001). The specificity increased from 87.9% in phase I to 95.6% in phase II, which helps to avoid over referrals (difference: +7.7%, 95% CI +1.4% to +13.9%; p=0.069). The lowest improvement in sensitivity for detecting any DR was observed between level 3 graders in Vietnam

**Table 1** Diagnostic test accuracy of DR graders in Vietnam against a reference standard from the UK, including ungradable images

| | Level 1 graders (n=410 patient images) | Level 2 graders (n=410 patient images) | Level 3 graders (n=260 patient images) |
|---|---|---|---|
| **Any DR** | | | |
| Sensitivity (%) (95% CI) | 41.8 (33.9 to 50.1) | 42.5 (34.5 to 50.7) | 42.2 (33.1 to 51.8) |
| Specificity (%) (95% CI) | 87.9 (83.3 to 91.7) | 98.8 (96.6 to 99.8) | 100 (97.5 to 100) |
| PPV (%) (95% CI) | 67.4 (57.0 to 76.6) | 95.6 (87.6 to 99.1) | 100 (92.7 to 100) |
| NPV (%) (95% CI) | 71.7 (66.4 to 76.7) | 74.3 (69.3 to 78.8) | 68.2 (61.5 to 74.5) |
| **Referable DR** | | | |
| Sensitivity (%) (95% CI) | 19.2 (9.63 to 32.5) | 13.5 (5.59 to 25.8) | 10.5 (2.94 to 24.8) |
| Specificity (%) (95% CI) | 97.2 (94.9 to 98.7) | 100 (99.0 to 100) | 99.5 (97.5 to 100) |
| PPV (%) (95% CI) | 50.0 (27.2 to 72.8) | 100 (59.0 to 100) | 80.0 (28.4 to 99.5) |
| NPV (%) (95% CI) | 89.2 (85.7 to 92.1) | 88.8 (85.3 to 91.7) | 86.7 (81.9 to 90.6) |
| **Referable DMO** | | | |
| Sensitivity (%) (95% CI) | 5.8 (1.91 to 13.0) | 20.9 (12.9 to 31.0) | 16.9 (8.76 to 28.3) |
| Specificity (%) (95% CI) | 97.2 (94.8 to 98.7) | 99.4 (97.8 to 99.9) | 100 (98.1 to 100) |
| PPV (%) (95% CI) | 35.7 (12.8 to 64.9) | 90.0 (68.3 to 98.8) | 100 (71.5 to 100) |
| NPV (%) (95% CI) | 79.5 (75.2 to 83.4) | 82.6 (78.4 to 86.2) | 78.3 (72.7 to 83.3) |

Grading criteria: UK National Diabetic Eye Screening Programme (NDESP) classification system (see online supplemental table S1 for more details).
Any DR, is defined as grades R1, R2, R3s, R3a and U. Referable DR is defined as grades R2, R3a and U. Referable DMO is defined as grades M1 and U.
Sensitivity is the ability of a test to correctly identify patients with a disease and specificity is the ability of a test to correctly identify people without the disease Positive predictive value (PPV) is the proportion of those who test positive who have the condition (true positives) and negative predictive value (NPV) is the proportion of those who test negative who do not have the condition (true negatives).
DMO, diabetic macular oedema; DR, diabetic retinopathy.

(difference: +9.1%, 95% CI −9.0% to +27.1%; p=0.326), although their specificity remained 100% at phase I and phase II. There was a significant improvement in sensitivity for detecting DR and referable DMO at all grader levels: sensitivities after training ranged between 40% and 61.5% (table 2). Further improvement in sensitivity was observed when ungradable images were excluded from the analysis in phase II: sensitivities ranged from 55.6 to 97.6% for any DR, 55.6%–88.9% for referable DR and 60.0–90.0% for referable DMO (online supplemental table S7). The overall prevalence of DR in this study can be found in online supplemental table S8.

## DISCUSSION

Results from our study demonstrated poor sensitivity and specificity for detecting all levels of DR, especially referable DR, in the early stages of programme delivery. This translates into increased costs to the healthcare system due to missed opportunities for early treatment and unnecessary examinations for false-positive referrals. The quality of patient care also suffers. Didactic DR training was delivered to graders in Vietnam over a 2-year period by trained DR graders from the UK and Vietnam. Training was specifically targeted to address problems identified in the phase I testing,[10] and quality control testing using international test and training (iTAT) were also undertaken. The iTAT is an online platform offering monthly quality assurance and training for graders who work in DR screening. It is a useful platform for graders to improve their skills in the detection of DR from ophthalmic images. In the UK, it is compulsory for graders to complete monthly test sets (each set consisting of 20 retinal images with a range of DR severities). If graders fall below the agreed threshold, additional training and support is provided.[7] This study demonstrates that these steps led to improved grading accuracy for all classes of patients and graders. We found that the main discordance between graders lay in their ability to detect ungradable images; therefore, targeted training must be given to ensure such patients are referred to the next level (slit-lamp examination).

According to the UK National Institute for Clinical Excellence guidelines, DR screening tests must have at least 80% sensitivity and 95% specificity with a technical failure of 5% or less.[11] These requirements were not met here for sensitivity but may not be applicable to LMICs. Results can be poor in low-resource settings for a variety of reasons, including higher prevalence of unoperated lens opacity impacting clarity of photographs, use of nurses rather than professional photographers for image capture

**Table 2** Diagnostic test accuracy of DR graders in Vietnam against a reference standard from the UK after additional DR training was delivered

| | Level 1 graders (n=115 patient images) | Level 2 graders (n=115 patient images | Level 3 graders (n=61 patient images) |
|---|---|---|---|
| **Any DR** | | | |
| Sensitivity (%) (95% CI) | 87.2 (74.3 to 95.2) | 68.1 (52.9 to 80.9) | 51.3 (34.8 to 67.6) |
| Specificity (%) (95% CI) | 95.6 (87.6 to 99.1) | 95.6 (87.6 to 99.1) | 100 (84.6 to 100) |
| PPV (%) (95% CI) | 93.2 (81.3 to 98.6) | 91.4 (76.9 to 98.2) | 100 (83.2 to 100) |
| NPV (%) (95% CI) | 91.5 (82.5 to 96.8) | 81.3 (71.0 to 89.1) | 53.7 (37.4 to 69.3) |
| P-value comparing sensitivity to Phase I | p=0.000 | p=0.002 | p=0.326 |
| **Referable DR** | | | |
| Sensitivity (%) (95% CI) | 53.3 (26.6 to 78.7) | 40.0 (16.3 to 67.7) | 58.3 (27.7 to 84.8) |
| Specificity (%) (95% CI) | 90.0 (82.4 to 95.1) | 93.0 (86.1 to 97.1) | 100 (92.7 to 100) |
| PPV (%) (95% CI) | 44.4 (21.5 to 69.2) | 46.2 (19.2 to 74.9) | 100 (59.0 to 100) |
| NPV (%) (95% CI) | 92.8 (85.7 to 97.0) | 91.2 (83 to 95.9) | 90.7 (79.7 to 96.9) |
| P-value comparing sensitivity to Phase I | p=0.009 | p=0.022 | p=0.001 |
| **Referable DMO** | | | |
| Sensitivity (%) (95% CI) | 56.3 (29.9 to 80.2) | 43.8 (19.8 to 70.1) | 61.5 (31.6 to 86.1) |
| Specificity (%) (95% CI) | 97.0 (91.4 to 99.4) | 93.9 (87.3 to 97.7) | 100 (92.6 to 100) |
| PPV (%) (95% CI) | 75.0 (42.8 to 94.5) | 53.8 (25.1 to 80.8) | 100 (63.1 to 100) |
| NPV (%) (95% CI) | 93.2 (86.5 to 97.2) | 91.2 (83.9 to 95.9) | 90.6 (79.3 to 96.9) |
| P-value comparing sensitivity to Phase I | p=0.000 | p=0.051 | p=0.002 |

Grading criteria: UK National Diabetic Eye Screening Programme (NDESP) classification system (See online supplemental table S1 for more details).
Criteria: any DR is defined as grades R1, R2, R3s, R3a and U. Referable DR is defined as grades R2, R3a and U. Referable DMO is defined as grades M1 and U. $\chi^2$ used to compare sensitivity between phase I and II.
Sensitivity is the ability of a test to correctly identify patients with a disease, and specificity is the ability of a test to correctly identify people without the disease. Positive predictive value (PPV) is the proportion of those who test positive who have the condition (true positives), and negative predictive value (NPV) is the proportion of those who test negative who do not have the condition (true negatives).
DMO, diabetic macular oedema; DR, diabetic retinopathy.

and poor compliance with photography among patients who have not previously undergone such examinations.

Quality assessment in such settings is crucial, and programmatic changes based on models such as the UK DESP can be successful in enhancing grader accuracy in LMICs settings. However, it is important for countries to adapt their own DR classification system and referral pathways to meet their own requirements. As an example, the UK system (England, Wales and Northern Ireland) use the grade M0 for no maculopathy and M1 for referable maculopathy. In Scotland, M0 denotes no maculopathy, M1 observable maculopathy and M2 referable maculopathy allowing some monitoring of maculopathy to take place at screening level. This reduces the burden on the hospital system. The implication for LMICs is that being aware of hospital capacity at the planning stage might mean that they need to safely adapt an accepted grading system to their needs. Most importantly, the role of affiliated hospitals (and partnerships, coordination among training institutions and practical hospitals) are crucial for DR grading quality improvement.

Studies in LMICs have assessed the accuracy of non-medical graders and medical graders in the detection of DR and found that both grader types are capable of achieving moderate to high sensitivity for detecting DR.[12–15] Comparable with our findings, a study in China found that non-medical DR graders achieved higher sensitivity (0.82–0.94%) and specificity (0.91–0.98%) compared with rural ophthalmologists (sensitivity=0.65–0.95%, specificity=0.59–0.95%).[16] In DR screening, it is vital to detect referable and sight-threatening DR (STDR) to prevent blindness, but it is equally important to detect normal cases to prevent unnecessary referrals to already overburdened hospital clinics. Screening provides an opportunity for graders to discuss with patients the importance of managing diabetes to reduce the risk of visual impairment from DR.

Some studies have described what training interventions were used to train their graders, and key elements may be incorporated into our training programme in the future.[15 17 18] In the UK, the DR grading course by the Gloucestershire Retinal Education Group is required for grader certication. The high costs of this course may be more challenging in LMICs due to limited funding.[6]

## Strengths

This study describes the impact of a training intervention to improve the quality of DR grading in an LMIC. The inclusion of ungradable images in this study was a logical decision, particularly when the prevalence of cataract (which often renders DR images ungradable) is high in LMICs.[19] Dense cataracts normally obstruct the view of the fundus, making it difficult to obtain clear fundus photographs and assign a DR grade. In these instances, referring patients to an eye clinic for further assessment and treatment as needed is required. Determining sensitivity and specificity at the patient level is also important from a DESP implementation perspective. In the UK and Vietnam, both eyes are typically examined for DR, and a single outcome is assigned to the patient, as was done here. For these reasons, we feel our analytic approach, and thus results, are relevant to these settings.

## Limitations

Limitations for this study have also been acknowledged. Data from this study represent routine clinical practice. In daily DR screening, not all patients undergoing primary (level 1) and secondary (level 2) grading proceed to arbitration grading (level 3). This means a proportion of images were not graded by arbitration graders as outlined in figure 1 and figure 2. Second, only 6/14 graders from phase I were included in phase II grading; however, the distribution of grader levels was similar. Third, though the proportion of patients excluded was small, we are unable to fully characterise the reasons for these exclusions, due to the nature of the study as a programmatic evaluation. Some potential reasons for this are a patient's unwillingness to participate in the study, graders having forgotten to ask for patient consent to participate in the study and patient inability to comply with image capture. Fourth, pupil status (size and cataract status) was not recorded in this study, and this can be important for LMICs. Finally, it was not practical for the UK reference standard to examine patients clinically in Vietnam; however, the method of grading by a certified DR grader or clinical specialist is widely used as the reference standard in many screening programmes.

## CONCLUSIONS

This paper shows how grading accuracy was particularly low among all grader groups in Vietnam in the first 6 months of DESP implementation. Many factors may have contributed to poor grader performance, including inadequate training and feedback, insufficient time to participate in quality assurance testing and competing work responsibilities. After additional training, testing and quality assurance systems were implemented in Vietnam, DTA improved among all grader groups; however, more work is still needed. In particular, training graders to detect ungradable cases is crucial. With continuous quality improvement, monthly iTAT, periodic DR workshops and review of certification, we would expect the DR sensitivity and specificity to improve further. A qualitative study to determine why the initial training intervention was less successful should be explored. Since further improvements are required, understanding how other countries implement such programmes would be beneficial. Future studies should outline what DR training interventions were used, state relevant training courses and explain what quality assurance measures are in place. The findings from this study are important for DESP programme planners in Vietnam and other LMICs, highlighting the importance of quality monitoring and directed retraining as needed.

Artificial intelligence (AI) is likely to significantly change future approaches to DR grading. Continued attention to maximising accuracy of human graders is still highly relevant today, especially for low-resource settings, as AI systems must be validated locally against a gold standard of proven expert human graders. Differences between the high-quality images used to train most existing AI systems and the types of images encountered in low-resource settings, with high rates of prevalent lens opacity, less-well-trained photographers and lower cost cameras, mean that such validation must almost certainly occur at the local level. The continued importance of reliable human graders in low-resource settings is further underscored by the fact that few systems are able to function fully autonomously without input from existing graders.

**Author affiliations**

[1]Centre of Public Health, Queen's University Belfast School of Medicine Dentistry and Biomedical Sciences, Belfast, UK
[2]Zhongshan Ophthalmic Center, Sun Yat-sen University, Guangzhou, China
[3]ORBIS International, New York, New York, USA
[4]Department of Ophthalmology, Hanoi Medical University, Hanoi, Viet Nam
[5]Save Sight Institute, The University of Sydney School of Medicine, Sydney, New South Wales, Australia
[6]Eye Hospital of Wenzhou Medical University, Wenzhou, Zhejiang Province, China
[7]Orbis International in Vietnam, Hanoi, Viet Nam
[8]Vitreo-Retina Department, Ho Chi Minh Eye Hospital, Ho Chi Minh City, Viet Nam
[9]Department of Ophthalmology, Belfast Health and Social Care Trust, Belfast, UK

**Acknowledgements** The authors are very grateful to all participants in Vietnam and Northern Ireland who generously gave their time for this study and to Orbis International.

**Contributors** KC accepts full responsibility for the work and/or the conduct of the study, had access to the data, and controlled the decision to publish. All authors have made substantial contributions to the conception of the design of the work. KC and GV analysed and interpreted the data. All authors read and approved the final manuscript.

**Funding** The project was funded by the Department for the Economy (DfE) – Global Challenges Research Fund (GCRF) Awards (Grant number: DFEGCRF17-18/

Peto). None of the funders were involved in the design or conduct of the study; preparation, review or approval of the manuscript; or decision to submit the manuscript for publication. NC is supported by the Ulverscroft Foundation (UK) (Grant number: N/A). KC is funded by the Wellcome Trust (Grant number: 204835/Z/16/A).

**Competing interests**  NC is employed as a Research Director by Orbis International.

**Patient and public involvement**  Patients and/or the public were not involved in the design, or conduct, or reporting, or dissemination plans of this research.

**Patient consent for publication**  Not applicable.

**Ethics approval**  This study involves human participants and was approved by an ethics committee or institutional board. This research adhered to the tenets of the Declaration of Helsinki. Ethical approval was granted by the Hanoi Medical University Institutional Review Board in Bio-Medical Research, Vietnam (No. 0518/HMU IRB). Written informed consent was obtained from all participants prior to their being interviewed. Participants gave informed consent to participate in the study before taking part.

**Provenance and peer review**  Not commissioned; externally peer reviewed.

**Data availability statement**  All data relevant to the study are included in the article or uploaded as supplementary information.

**ORCID iDs**
Katie Curran http://orcid.org/0000-0002-7071-109X
Gianni Virgili http://orcid.org/0000-0002-9960-2989
Prabhath Piyasena http://orcid.org/0000-0002-0236-0101
Tunde Peto http://orcid.org/0000-0001-6265-0381

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
