## [Reviewer comments · BMJ Open]

ARTICLE DETAILS

TITLE (PROVISIONAL)	The impact of targeted diabetic retinopathy training for graders in Vietnam and the implications for future diabetic retinopathy screening programmes: a diagnostic test accuracy study
AUTHORS	Curran, Katie; Congdon, Nathan; Hoang, Tung; Lohfeld, Lynne; Nguyen, Van; Nguyen, Hue; Nguyen, Quan; Dardis, Catherine; Virgili, Gianni; Piyasena, Prabhath; Tran, Huong; Salongcay, Recivall; Tung, Mai; Peto, Tunde

VERSION 1 – REVIEW

REVIEWER	Rani, Padmaja Kumari Univ Alabama Birmingham
REVIEW RETURNED	08-Dec-2021

GENERAL COMMENTS	-The study involves an important and relevant topic. - The distribution of image dataset - regarding quality, diagnosis of DR should be presented - One should calculate sensitivity and specificity after removing ungradable images - Ungradable images were referred to next level of care --hence one should assess whether graders could refer them to next level? this point is not clear - What was the DR training (curriculum and duration details) imparted for graders -- because present study results show suboptimal performance. This is a surprising finding. Even ophthalmologists (level 2) and senior ophthalmologists (level 3) could not achieve adequate sensitivity and specificities for DR. This finding is not convincing. Need more in-depth understanding of training curriculum to understand above suboptimal performance Authors report that their data on training interventions is first reported from LMIC, this is not correct. We have many studies reporting DR training interventions from Bangladesh, Nepal, Sri Lanka, Pakistan, Zanzibar, India with adequate sensitivities and specificities for detection of DR as below Authors can review these articles and discuss in the discussion 1. Thapa R, Bajimaya S, Pradhan E, Sharma S, Kshetri B, Paudyal G. Agreement on Grading Retinal Findings of Patients with Diabetes Using Fundus Photographs by Allied Medical Personnel when Compared to an Ophthalmologist at a Diabetic Retinopathy Screening Program in Nepal. Clin Ophthalmol 2020.14:2731-2737. 2. Rani PK, Peguda HK, Chandrashekhher M, Swarna S, Jonnadula GB, James J et al. Capacity building for diabetic retinopathy screening by optometrists in India: Model description and pilot results. Indian J Ophthalmol 2021.69:655-659. 3. Ramakrishnan R, Abdul Khadar SM, Srinivasan K, Kumar H,
--

	Vijayakumar V. Diabetes mellitus in the Tamil Nadu State-Noncommunicable diseases nurse model in diabetic retinopathy screening. Indian J Ophthalmol 2020.68:S78-S82. 4.Omar FJ, Sheeladevi S, Rani PK, Ning G, Kabona G. Evaluating the effectiveness of opportunistic eye screening model for people with diabetes attending diabetes clinic at Mnazi Mmoja Hospital, Zanzibar. BMC Ophthalmol 2014.14:81. 5.Shah M, Noor A, Ormsby GM, Islam FA, Harper CA, Keeffe JE. Task sharing: Development of evidence-based co-management strategy model for screening, detection, and management of diabetic retinopathy. Int J Health Plann Manage 2018.33:e1088-e1099. 6.Srinivasan S, Shetty S, Natarajan V, Sharma T, Raman R. Development and Validation of a Diabetic Retinopathy Referral Algorithm Based on Single-Field Fundus Photography. PLoS One 2016.11:e0163108.
--	--

REVIEWER	Salamanca, Omar ORBIS International, Ophthalmology
REVIEW RETURNED	31-Jan-2022

GENERAL COMMENTS	The paper presents the results of training in DR imaging grading in Vietnam. The subject is very important and necessary, therefore the results of this type of study are very necessary and welcome. However, the text does have some points that deserve attention from the authors. The abstract needs to be reviewed and must be structured. A guide about this can be found here: "Cohen JF, Korevaar DA, Gatsonis CA, Glasziou PP, Hooft L, Moher D, Reitsma JB, de Vet HC, Bossuyt PM; STARD Group. STARD for Abstracts: essential items for reporting diagnostic accuracy studies in journal or conference abstracts. BMJ. 2017 Aug 17;358:j3751. doi: 10.1136/bmj.j3751. PMID: 28819063. " It is recommended to include the type of study and the number of subjects/participants included (images reviewed/graded) Training characteristics for DR images grading should be widely described in terms of hours, objectives, evaluation, and other aspects. Is it theoretical, theoretical-practical? Is any kind of certificate issued? Did all participants attend most of the time? Is this training similar to the one in the UK? A better description would help the reader to understand this process. Training in the use of cameras for DR images is mentioned. Was this for the technical level? Level 2 DR graders and level 3 DR graders have a difference in terms of clinical experience, but apparently have the same level of training in DR imaging grading. Is this a factor that modifies/alters the results? In phase 2 of the study, a subgroup of 6 of 14 graders were included. What was the reason for not including all the previous participants? Are these subjects somehow "representative" of the initial group?
---

	Level 1 DR graders group is very heterogeneous in terms of training and background. What implications might this have for this training results? Although this is known to those familiar with DR screening, these terms should be described: “Any DR, is defined as grades R1, R2, R3s, R3a and U.” At the beginning of the results section, a brief description of the number of subjects examined and related relevant information could be presented, as a courtesy to the reader. This information can be inferred from the graphs but could be included there. In the results section, the paragraphs on lines 185-187 and 197-198 might fit better in the discussion section. In lines 99 -100: should it be Level 2 DR graders and level 3 DR graders? Line 138: “Ungradable images were referred for further slit-lamp examination” may be changed to “participants with ungradable images were referred for further.....” Line 145 it is redundant, there is information that has already been mentioned previously in the text. These paragraphs seem to be contradictory, please review: "To date, there is insufficient evidence on DR grading accuracy using non- mydriatic digital imaging by trained graders in LMICs, and even less about the capacity of 92 DESPs in LMICs to improve where poor accuracy is detected." (Lines 90-92) "Studies in LMICs and HICs have assessed the accuracy of non-medical graders and medical graders in the detection of DR and found that both grader types are capable of achieving moderate-high sensitivity for detecting DR." (Lines 218-220) Please justify this statement: "This formal training qualification and continuous monitoring and evaluation are crucial to achieve optimal sensitivity, which may be more challenging in terms of costs and capacity for LMICs." (Lines 224-226)
--	--

REVIEWER	Woodward, Richmond Duke University, Ophthalmology
REVIEW RETURNED	07-Feb-2022

GENERAL COMMENTS	The manuscript, “The impact of targeted diabetic retinopathy training for graders in Vietnam and the implications for future diabetic retinopathy screening programmes” expands on previous studies of local non-ophthalmology diabetic retinopathy (DR) graders in LMICs by studying not only baseline sensitivity and specificity of grading, but also measuring improvement following a targeted teaching intervention. This study is important given the rising burden of DR in LMICs worldwide and the likelihood that widespread implementation of artificial intelligence in DR screening is still years away. Below are comments and suggested minor revisions:
---

	Gaps:  1.The abstract does not inform on the stage of DR being graded, all stages, just referable, or both. Some clarity might be helpful. 2.A brief mention of how the Level 1 graders were initially trained would be helpful. Reference 7 takes the reader to an on-line course offered by the Gloucestershire Retinal Education Group. Is this similar to the program taught to the Level 1 graders? Did the teaching include a practical exam or any kind of exam? It seems important to know what the expected level of training was for the Level 1 graders if this study was ever to be duplicated, or if other programs in LIMCs aimed to create their own teaching programs based on this study. The maximal impact of targeted diabetic retinopathy grading can be met only by teaching non-ophthalmologist Level 1 graders to interpret retinal images for DR and DMO. 3.The authors conclude the results of sensitivity and specificity after initial teaching and then re-teaching are not satisfactory. However, they do not indicate what level of results is satisfactory. The authors indicate they compare results against the reference standard in the UK, but do not state what that standard is. If the results of the Vietnamese graders is compared to standards in the UK, is that a reasonable goal or expectation? While the aim of the study was not to determine what the target goal for sensitivity and specificity should be, elaborating and discussing this puts the study results in a real-world context. For example, see line 192 - “sensitivities after training were still insufficient and comprised between about 40% and 61.5%” without mentioning what the desired sensitivity is - the reader is left wondering. Similarly, line 210 states, “results remain suboptimal for a screening programme” without explaining what optimal is. 4.An appendix at the end describing the UK grading scale in more detail would be informative for readers not using the UK grading system. The paper lists the scale using letters, which is not meaningful to readers unfamiliar with the UK system. Table 1 would benefit by referencing an appendix at the end of the paper briefly summarizing the National Screening Committee (NSC) classification system beyond just the letters, R1, R2, R3, etc. While there is a reference to the UK grading system, I believe a short description of the grading system would increase the impact of the paper. 5.Certain areas are limited by generality and could use more detail, e.g. line 115 “quality control methods” are not described or referenced. Possible overreach or confusing use of terminology: Line 214-215: “It is fundamental for countries to adapt their own DR classification system and referral pathways...” This statement is not backed up with any explanation by what is meant to “adapt” a DR classification system. I can understand adapting a referral pattern but was surprised by the notion of adapting a classification system. Can the authors please elaborate and/or give examples where this is currently done world-wide? Other specific comments: Line 100 - typo, “Level 2 DR graders” should read “Level 3 DR graders” line 123 - is the fundus camera handheld or tabletop line 147 - what were the specific areas or problems that required retraining? Line 159 - it would be helpful to describe what sensitivity and specificity refer to as all readers might not
--	---

	be familiar with this line 205 - "training over a two-year period" Previously, two teaching sessions between the first grading and second grading were described. It is unclear if the training over a two-year period refers to more than two specific teaching sessions or included ongoing training during this time. Line 149 states, "Additionally, regular testing and training for quality assurance purposes was also introduced, similar to UK DESP models." The statement in line 149 is not referenced and a brief description of what regular testing and training entailed, or at minimum including a reference would be helpful. This is important so that countries or programs aiming to implement a similar system know what was done to provide the gains in grading retinal images documented in this study. This also loops back to the comment above asking for a brief description of the training of Level 1 DR graders initially, as sharing information about this may help others planning their own training programs. Line 248 - pupil status, what specifically about pupil status is lacking
--	---

VERSION 1 – AUTHOR RESPONSE

Reviewer: 1

Dr. Padmaja Kumari Rani, Univ Alabama Birmingham

Comments to the Author:

The study involves an important and relevant topic.

#5 The distribution of image dataset - regarding quality, diagnosis of DR should be presented

Thank you for your comment. These figures can be found in Table S4 of the supplementary material.

#6 One should calculate sensitivity and specificity after removing ungradable images

Thank you for your comment. We have included tables in the supplementary material after removing ungradable images (Tables S5 and S6).

#7 Ungradable images were referred to next level of care --hence one should assess whether graders could refer them to next level ? this point is not clear

We found that the main discordance was the graders' ability to detect ungradable images, therefore, targeted training must be given to ensure patients are referred to the next level (slit-lamp examination) (Line 223-225).

#8 What was the DR training (curriculum and duration details) imparted for graders -- because present study results show suboptimal performance. This is surprising finding. Even ophthalmologists (level 2) and senior ophthalmologists (level 3) could not achieve adequate sensitivity and specificities for DR. This finding is not convincing. Need more in-depth understanding of training curriculum to understand above suboptimal performance.

Thank you. A team of five Vietnamese doctors and medical administrators visited a Northern Ireland (NI) DESP in September 2017 to receive screener/grader training and training in administration and failsafe methods. These were then incorporated into the Vietnamese DESP's protocols, and processes and screening started in December 2017 in five locations in Vietnam. In January/February 2018, a senior ophthalmic nurse and an optometrist from the Belfast Trust, both certified and trained

in DR grading visited Vietnam to deliver refresher training to graders involved in the DESPs. During this time, we responded to any queries and reinforced the importance of taking good quality fundus images. Grader's training focused on ocular anatomy, retinal disease, DR signs, DR grading (based on the UK DESP grading classification system) and appropriate referral pathways and management. This training was delivered in conjunction with Orbis Vietnam. All graders were provided with their own grading manual on DR screening and grading. Each grader was assigned a certificate from Orbis also. Orbis played a leading role in providing continuous education and support for graders. In 2019, the UK graders visited Vietnam again to provide additional DR training. A flow chart of the training is outlined below and an agenda outlining the specific topics covered in the training sessions is outlined in the supplementary material.

#9 Authors report that their data of training intervention is first reported from LMIC, this is not correct.

We have many studies reporting DR training interventions from Bangladesh, Nepal, Srilanka, Pakistan, zanzibar India with adequate sensitivities and specificities for detection of DR as below

Authors can review these articles and discuss in the discussion:

1. Thapa R, Bajimaya S, Pradhan E, Sharma S, Kshetri B, Paudyal G. Agreement on Grading Retinal Findings of Patients with Diabetes Using Fundus Photographs by Allied Medical Personnel when Compared to an Ophthalmologist at a Diabetic Retinopathy Screening Program in Nepal. *Clin Ophthalmol* 2020.14:2731-2737.
2. Rani PK, Peguda HK, Chandrasheker M, Swarna S, Jonnadula GB, James J et al. Capacity building for diabetic retinopathy screening by optometrists in India: Model description and pilot results. *Indian J Ophthalmol* 2021.69:655-659.
3. Ramakrishnan R, Abdul Khadar SM, Srinivasan K, Kumar H, Vijayakumar V. Diabetes mellitus in the Tamil Nadu State-Noncommunicable diseases nurse model in diabetic retinopathy screening. *Indian J Ophthalmol* 2020.68:S78-S82.
4. Omar FJ, Sheeladevi S, Rani PK, Ning G, Kabona G. Evaluating the effectiveness of opportunistic eye screening model for people with diabetes attending diabetes clinic at Mnazi Mmoja Hospital, Zanzibar. *BMC Ophthalmol* 2014.14:81.
5. Shah M, Noor A, Ormsby GM, Islam FA, Harper CA, Keeffe JE. Task sharing: Development of evidence-based co-management strategy model for screening, detection, and management of diabetic retinopathy. *Int J Health Plann Manage* 2018.33:e1088-e1099.
6. Srinivasan S, Shetty S, Natarajan V, Sharma T, Raman R. Development and Validation of a Diabetic Retinopathy Referral Algorithm Based on Single-Field Fundus Photography. *PLoS One* 2016.11:e0163108.

Thank you very much for providing me with these relevant references. I have now included some of the above references in my discussion. I have also included our training model in the supplementary material. A study in South India showed an acceptable level of sensitivity and specificity among optometrists for the presence of either sight-threatening or non-sight-threatening DR and DMO. They outlined their training methodology, and some elements may be employed into our training in the future. <https://www.ncbi.nlm.nih.gov/pmc/articles/PMC7942067/pdf/IJO-69-655.pdf>

Reviewer: 2

Dr. Omar Salamanca, ORBIS International, Universidad del Valle

Comments to the Author:

The paper presents the results of training in DR imaging grading in Vietnam. The subject is very important and necessary, therefore the results of this type of study are very necessary and welcome. However, the text does have some points that deserve attention from the authors.

#10 The abstract needs to be reviewed and must be structured. A guide about this can be found here: "Cohen JF, Korevaar DA, Gatsonis CA, Glasziou PP, Hooft L, Moher D, Reitsma JB, de Vet HC, Bossuyt PM; STARD Group. STARD for Abstracts: essential items for reporting diagnostic accuracy studies in journal or conference abstracts. *BMJ*. 2017 Aug 17;358:j3751. doi: 10.1136/bmj.j3751. PMID: 28819063. " It is recommended to include the type of study and the number of subjects/participants included (images reviewed/graded)

Thank you for your comments, the abstract has been amended. The details can also be found in figure 1 and figure 2.

#11 Training characteristics for DR images grading should be widely described in terms of hours, objectives, evaluation, and other aspects. Is it theoretical, theoretical-practical? Is any kind of certificate issued? Did all participants attend most of the time? Is this training similar to the one in the UK? A better description would help the reader to understand this process.

See comment #8 above.

#12 Training in the use of cameras for DR images is mentioned. Was this for the technical level?

Technicians and nurses were trained to take good quality fundus photographs and anterior photographs.

#13 Level 2 DR graders and level 3 DR graders have a difference in terms of clinical experience, but apparently have the same level of training in DR imaging grading. Is this a factor that modifies/alters the results?

Reviewing patients in clinical practice is often different to grading fundus images for DR severity. Whilst the DR features will be familiar to those with more clinical practice, graders should take their time, grade images in a systematic way and make use of all available filters such as the red-free filter. This was highlighted in our previous study <https://www.nature.com/articles/s41433-021-01554-6.pdf>. In some case, graders in training may be more open to learning new concepts compared with practicing graders who may be biased towards concepts acquired with experience. This was found in a study by Srinivasan et al, <https://journals.plos.org/plosone/article?id=10.1371/journal.pone.0163108>

#14 In phase 2 of the study, a subgroup of 6 of 14 graders were included. What was the reason for not including all the previous participants? Are these subjects somehow “representative” of the initial group?

Yes, the subjects are representative of the initial group and the reason they were not included again was because they moved from this programme of work.

#15 Level 1 DR graders group is very heterogeneous in terms of training and background. What implications might this have for this training results?

All level 1 graders received the same training; therefore, this should not have any implications on their training results. For example, people from many backgrounds (medical or non-medical) are trained to be involved in the UK DESP model provided they undertake specific training and certification. Some of the certified non-medical graders in the UK DESP are top quality graders, even without prior knowledge of DR.

#16 Although this is known to those familiar with DR screening, these terms should be described: “Any DR, is defined as grades R1, R2, R3s, R3a and U.”

Thank you, I have now defined these more clearly in the supplementary material, Table S1. DR classifications: R1 is background retinopathy, R2 is pre-proliferative diabetic retinopathy, R3s is stable proliferative diabetic retinopathy, R3a is active proliferative diabetic retinopathy and U is ungradable images.

#17 At the beginning of the results section, a brief description of the number of subjects examined and related relevant information could be presented, as a courtesy to the reader. This information can be inferred from the graphs but could be included there.

Fundus images were graded for 412 patients in phase I and 115 patients in phase II (Figure 1 and figure 2).

#18 In the results section, the paragraphs on lines 185-187 and 197-198 might fit better in the discussion section.

Thank you for your suggestions. I agree that lines 185-187 would fit better into the discussion; however, we feel it is important to keep them in the results section to provide context to the remaining lines within that paragraph. Line 197-198 was repeated in the discussion; therefore, I have removed it from the results. (lines 119-120)

#19 In lines 99 -100: should it be Level 2 DR graders and level 3 DR graders?

Yes, thank you for identifying this error.

#20 Line 138: "Ungradable images were referred for further slit-lamp examination" may be changed to "participants with ungradable images were referred for further....."

Thank you, I have made this change as requested. (lines 143-144)

#21 Line 145 it is redundant, there is information that has already been mentioned previously in the text.

Thank you for your comment, I have now removed this text

#22 These paragraphs seem to be contradictory, please review:

"To date, there is insufficient evidence on DR grading accuracy using non- mydriatic digital imaging by trained graders in LMICs, and even less about the capacity of DESPs in LMICs to improve where poor accuracy is detected." (Lines 90-92)

"Studies in LMICs and HICs have assessed the accuracy of non-medical graders and medical graders in the detection of DR and found that both grader types are capable of achieving moderate-high sensitivity for detecting DR." (Lines 218-220)

Thank you for highlighting this. I have now amended lines 90-92 to "To date, there is insufficient evidence on how DR grading accuracy using non-mydriatic digital imaging by trained graders in LMICs can be improved, where poor accuracy is detected". (Line 93-94).

#23 Please justify this statement: "This formal training qualification and continuous monitoring and evaluation are crucial to achieve optimal sensitivity, which may be more challenging in terms of costs and capacity for LMICs." (Lines 224-226)

Thank you, I have amended this paragraph. In the UK, the DR grading course by the Gloucestershire Retinal Education Group is required for grader certification. The high costs of this course may be more challenging for LMICs due to limited funding. (Lines 248-250)

Reviewer: 3

Dr. Richmond Woodward, Duke University

Comments to the Author:

Your work is vital to reducing the global burden of diabetic retinopathy. Please see the attached file.

The manuscript, "The impact of targeted diabetic retinopathy training for graders in Vietnam and the implications for future diabetic retinopathy screening programmes" expands on previous studies of local non-ophthalmology diabetic retinopathy (DR) graders in LMICs by studying not only baseline sensitivity and specificity of grading, but also measuring improvement following a targeted teaching intervention. This study is important given the rising burden of DR in LMICs worldwide and the likelihood that widespread implementation of artificial intelligence in DR screening is still years away.

Below are comments and suggested minor revisions:

#24 The abstract does not inform on the stage of DR being graded, all stages, just referable, or both. Some clarity might be helpful.

Thank you for your comment. I have now amended this paragraph;

In Phase I, the sensitivity for detecting **any DR** for all grader groups in Vietnam was low and improved in Phase II after additional training was delivered. The greatest improvement was seen among level 1 graders ($P < 0.001$) and the lowest improvement was observed among level 3 graders ($P = 0.326$). There was an improvement in sensitivity for detecting **any DR and referable diabetic macular oedema** between all grader levels and whilst the differences were statistically significant, the post-training values were suboptimal (41.8% to 61.5%). The main disagreement was the detection of ungradable images (lines 57-63).

#25 A brief mention of how the Level 1 graders were initially trained would be helpful. Reference 7 takes the reader to an on-line course offered by the Gloucestershire Retinal Education Group. Is this similar to the program taught to the Level 1 graders? Did the teaching include a practical exam or any kind of exam? It seems important to know what the expected level of training was for the Level 1 graders if this study was ever to be duplicated, or if other programs in LMICs aimed to create their own teaching programs based on this study. The maximal impact of targeted diabetic retinopathy grading can be met only by teaching non-ophthalmologist Level 1 graders to interpret retinal images for DR and DMO.

See comment #8

#26 The authors conclude the results of sensitivity and specificity after initial teaching and then re-teaching are not satisfactory. However, they do not indicate what level of results is satisfactory.

According to the UK National Institute for Clinical Excellence (NICE) guidelines, DR screening tests must have at least 80% sensitivity and 95% specificity with a technical failure of 5% or less. These requirements may not be applicable to LMICs, especially at the start of the programme where a relatively low number of patients are being screened (lines 225-229).

#27 The authors indicate they compare results against the reference standard in the UK, but do not stay what that standard is. If the results of the Vietnamese graders is compared to standards in the UK, is that a reasonable goal or expectation? While the aim of the study was not to determine what the target goal for sensitivity and specificity should be, elaborating and discussing this puts the study results in a real-world context. For example, see line 192 - "sensitivities after training were still insufficient and comprised between about 40% and 61.5%" without mentioning what the desired sensitivity is - the reader is left wondering. Similarly, line 210 states, "results remain suboptimal for a screening programme" without explaining what optimal is.

As per comment above

#28 An appendix at the end describing the UK grading scale in more detail would be informative for readers not using the UK grading system. The paper lists the scale using letters, which is not meaningful to readers unfamiliar with the UK system. Table 1 would benefit by referencing an appendix at the end of the paper briefly summarizing the National Screening Committee (NSC) classification system beyond just the letters, R1, R2, R3, etc. While there is a reference to the UK grading system, I believe a short description of the grading system would increase the impact of the paper.

Thank you for your comment, I have now included more details in the supplementary material.

#29 Certain areas are limited by generality and could use more detail, e.g., line 115 "quality control methods" are not described or referenced.

International Test and Training is an online platform offering monthly quality assurance and training for graders who work in DR screening. It is a useful platform for graders to improve their skills in the detection of DR from ophthalmic images. It is compulsory in the UK for graders to complete monthly test sets (each set consisting of 20 retinal images with a range of DR severities). If the graders fall below the agreed threshold, additional training and support is provided (216-221) https://assets.publishing.service.gov.uk/government/uploads/system/uploads/attachment_data/file/512832/The_Management_of_Grading.pdf.

<https://www.gregcourses.com/test-and-training>. <https://www.gov.uk/government/publications/diabetic-eye-screening-test-and-training-participation/diabetic-eye-screening-participation-in-the-grading-test-and-training-system>

Possible overreach or confusing use of terminology:

#30 Line 214-215: “It is fundamental for countries to adapt their own DR classification system and referral pathways...” This statement is not backed up with any explanation by what is meant to “adapt” a DR classification system. I can understand adapting a referral pattern but was surprised by the notion of adapting a classification system. Can the authors please elaborate and/or give examples where this is currently done world-wide

As an example, the UK system (England, Wales and Northern Ireland) uses the grade M0 for no maculopathy and M1 for referable maculopathy. In Scotland, M0 denotes no maculopathy, M1 observable maculopathy and M2 referable maculopathy allowing some monitoring of maculopathy to take place on screening level. This reduces the burden on the hospital system. The implication for LMICs is that being aware of hospital capacity at the planning stage might mean that they need to safely adapt an accepted grading system to their needs. (Lines 234-240).

#31 Line 100 - typo, “Level 2 DR graders” should read “Level 3 DR graders”

Thank you, this has now been amended.

#32 line 123 - is the fundus camera handheld or tabletop

Fundus images were taken using a tabletop fundus camera. I have now included this in line 128

#33 line 147 - what were the specific areas or problems that required retraining?

Thank you for your comment.

Targeted training must be given to ensure patients are referred to the next level (slit-lamp examination) (Lines 224-225) Specific areas or problems that required retraining have been outlined in our previous publication “Capturing the clinical decision-making processes of expert and novice diabetic retinal graders using a ‘think-aloud’ approach” <https://www.nature.com/articles/s41433-021-01554-6>.

#34 Line 159 - it would be helpful to describe what sensitivity and specificity refer to as all readers might not be familiar with this

Sensitivity is the ability of a test to correctly identify patients with a disease and specificity is the ability of a test to correctly identify people without the disease
Positive predictive value (PPV) is the proportion of those who test positive who have the condition (true positives) and negative predictive value (NPV) is the proportion of those who test negative who do not have the condition (true negatives). See definitions in Tables 1 and 2.

#35 line 205 - “training over a two-year period” Previously, two teaching sessions between the first grading and second grading were described. It is unclear if the training over a two-year

period refers to more than two specific teaching sessions or included ongoing training during this time.

Thank you for your comment. I have now included a training timeline in the supplementary material.

#36 Line 149 states, “Additionally, regular testing and training for quality assurance purposes was also introduced, similar to UK DESP models.” The statement in line 149 is not referenced and a brief description of what regular testing and training entailed, or at minimum including a reference would be helpful. This is important so that countries or programs aiming to implement a similar system know what was done to provide the gains in grading retinal images documented in this study. This also loops back to the comment above asking for a brief description of the training of Level 1 DR graders initially, as sharing information about this may help others planning their own training programs.

As explained in comment #29, International Test and Training is an online platform offering monthly quality assurance and training for graders who work in DR screening. It is a useful platform for graders to improve their skills in the detection of DR from ophthalmic images. It is compulsory in the UK for graders to complete monthly test sets (each set consisting of 20 retinal images with a range of DR severities). If the graders fall below the agreed threshold, additional training and support is provided. <https://www.gregcourses.com/test-and-training>. <https://www.gov.uk/government/publications/diabetic-eye-screening-test-and-training-participation/diabetic-eye-screening-participation-in-the-grading-test-and-training-system>

Graders in Vietnam were encouraged to engage regularly in iTAT and this was included in their training in the form of a practical session (see supplementary material).

#37 Line 248 - pupil status, what specifically about pupil status is lacking

Pupil status (pupil size, presence of cataract) was not recorded in this study and this can be important for LMICs. (Line 271).

VERSION 2 – REVIEW

REVIEWER	Rani, Padmaja Kumari LV Prasad Eye Institute
REVIEW RETURNED	18-May-2022

GENERAL COMMENTS	Authors should mention sensitivity and specificity values for detection of any DR and sight threatening DR in the abstract. Please present sensitivity and specificity values after excluding ungradable images.
--

REVIEWER	Salamanca, Omar ORBIS International, Ophthalmology
REVIEW RETURNED	28-May-2022

GENERAL COMMENTS	The authors have adequately addressed the comments of the reviewers.
--

REVIEWER	Woodward, Richmond
-----------------	--------------------

	Duke University, Ophthalmology
REVIEW RETURNED	01-Jun-2022

GENERAL COMMENTS	Your manuscript presents very important evidence that DR grading accuracy can be improved. My comments (attached) to the revised manuscript refer to putting the results in a greater context in the discussion if the word count permits, e.g. an expanded introduction with more background material and discussion with more context. This would include a brief discussion of should the standard of optimal be for any DR/DMO, or should the standard of what is optimal be solely for vision threatening DR/DMO as that is clinically most relevant. Throughout the manuscript the results for sensitivities after retraining are considered suboptimal, but for a LMIC can/should be this be put into greater context in terms of what other programs that have trained level 1 and 2 graders have found. I walked away from the manuscript asking myself, if after all the effort to train and retrain the graders and the accuracy is suboptimal, does the future lie in AI grading images?
--

VERSION 2 – AUTHOR RESPONSE

Reviewer: 1

Dr. Padmaja Kumari Rani, Univ Alabama Birmingham

Comments to the Author:

#2 Authors should mention sensitivity and specificity values for detection of any DR and sight threatening DR in the abstract.

Thank you, please see results section in abstract below:

Results: In Phase I, the sensitivity for detecting any DR for all grader groups in Vietnam was low (41.8-42.2%) and improved in Phase II after additional training was delivered (51.2-87.3%). The greatest improvement was seen among level 1 graders ($P < 0.001$) and the lowest improvement was observed among level 3 graders ($P = 0.326$). There was a statistically-significant improvement in sensitivity for detecting referable DR and referable diabetic macular oedema between all grader levels. The post-training values ranged from 40.0-61.5% (including ungradable images) and 55.6%-90.0% (excluding ungradable images). (Lines 60-66, page 4)

#3 Please present sensitivity and specificity values after excluding ungradable images.

Thank you, we have included these results in the manuscript and as tables in the supplementary material.

The sensitivity increased when ungradable images were excluded from the analysis, though it still remained low (47.9-50.8% for any DR; 22.2%-38.1% for referable DR and 9.3-26.5% for referable DMO) (Supplementary, Table S4). (Line 198-200, page 12)

Further improvement in sensitivity was observed when ungradable images were excluded from the analysis in Phase II: sensitivities ranged from 55.6 to 97.6% for any DR, 55.6%-88.9% for referable DR and 60.0-90.0% for referable DMO (Supplementary, Table S5). (Line 221-224, page 13)

Reviewer: 2

Dr. Omar Salamanca, ORBIS International, Universidad del Valle

Comments to the Author:

The authors have adequately addressed the comments of the reviewers.

Reviewer: 3

Dr. Richmond Woodward, Duke University

Comments to the Author:

I reviewed with great interest the revised manuscript, “The impact of targeted diabetic retinopathy training for graders in Vietnam and the implications for future diabetic retinopathy screening programmes”. I believe this study is very important given the rising burden of diabetic retinopathy in LMICs worldwide, and the great need to screen people for vision threatening diabetic retinopathy. Certain improvements are appreciated and add clarity, e.g., the new supplementary material.

#4 Please note on pg. 23 in the lower right and square, the order that the information is presented is not consistent with the other boxes in the flow chart.

Thank you, this amendment has been made.

#5 While the authors made improvements and clarified questions on concerns from the initial review, a few questions remain. My greatest concern is that the results are not presented in a greater context. Throughout the paper the results for sensitivity are compared to a “gold standard” in the UK and are found to be suboptimal. As part of the discussion, or as background earlier in the manuscript, it would be helpful to review what other studies in LMICs have found for sensitivity and specificity when non-professional graders are grading photos. Can the results from this study be placed in a greater context? There is a very brief discussion in line 224-248. Can the authors state what is optimal, in their opinion, for a LMIC, and in particular where should the bar be set for sensitivity for LMIC? Is it better to have photos screened at a suboptimal level then not screened at all? Rather than focus on sensitivity for detecting any DR, should the focus be only on sensitivity for detecting referable or sight threatening DR? Results for referable DR and DMO is presented in Table 2, which is very helpful. Yet again it would be important to put this in context.

Thank you for your valuable comments. We have included the following:

Studies in LMICs have assessed the accuracy of non-medical graders and medical graders in the detection of DR and found that both grader types are capable of achieving moderate-high sensitivity for detecting DR.[12-15] Comparable with our findings, a study in China found that non-medical DR graders achieved higher sensitivity (0.82-0.94%) and specificity (0.91-0.98%) compared to rural ophthalmologists (sensitivity=0.65-0.95%, specificity=0.59-0.95%) [16]. In DR screening, it is vital to detect referable and sight-threatening DR to prevent blindness, but it is equally important to detect normal cases to prevent unnecessary referrals to already overburdened hospital clinics. Screening provides an opportunity for graders to discuss with patients the importance of managing diabetes to reduce the risk of visual impairment from DR. (Line 286-295, page 16).

#6 Line 222-224 states accuracy of results improved but the “results remain suboptimal for a screening program”. This is a key statement that is not discussed further - please see above. In addition, there could be more discussion on the observation that the lowest improvement in sensitivity for detecting any DR was in level 3 graders. This implies that there is a ceiling for the graders with the most training and experience at baseline in interpreting retinal photos. Is this seen in other studies, how would the authors address this in the future?

See comment #5 above.

#7 In the conclusion, lines 287-289, the authors state other studies in the future should outline what DR training measures were used and what quality measures were taken. However, earlier in the paper there is not even cursory mention of specific studies from other countries that

have undertaken training of non-ophthalmologist graders so we can see how the Vietnam results fit in the bigger picture. I do not know if the authors are at the maximum word count, and possibly they cannot add any more background information. But if they are not at the maximum word count, I believe this would be very helpful/should be done.

Thank you for your comment. See comment #5 above.

#8 As noted in my comments to the authors, I walked away from the manuscript asking myself, if after all the effort to train and retrain the graders and the accuracy is suboptimal, does the future lie in AI grading images? I recommend accept after the authors determine if they can add more context as noted above.

Artificial intelligence (AI) is likely to significantly change future approaches to DR grading. Continued attention to maximising accuracy of human graders is still highly relevant today, especially for low-resource settings, as AI systems must be validated locally against a gold standard of proven expert human graders. Differences between the high-quality images used to train most existing AI systems and the types of images encountered in low-resource settings, with high rates of prevalent lens opacity, less-well-trained photographers, and lower-cost cameras, mean that such validation must almost certainly occur at the local level. The continued importance of reliable human graders in low-resource settings is further underscored by the fact that few systems are able to function fully autonomously without input from existing graders. (Line 364-378, page 18 and 19)